# Cancer-Specific Delivery of Proteolysis-Targeting Chimeras (PROTACs) and Their Application to Cancer Immunotherapy

**DOI:** 10.3390/pharmaceutics15020411

**Published:** 2023-01-26

**Authors:** Yujeong Moon, Seong Ik Jeon, Man Kyu Shim, Kwangmeyung Kim

**Affiliations:** 1Department of Bioengineering, Korea University, Seoul 02841, Republic of Korea; 2Biomedical Research Institute, Korea Institute of Science and Technology (KIST), Seoul 02792, Republic of Korea; 3Graduate School of Pharmaceutical Sciences, College of Pharmacy, Ewha Woman’s University, Seoul 03760, Republic of Korea

**Keywords:** proteolysis-targeting chimera (PROTAC), protein degradation, drug delivery system, cancer-targeted therapy, cancer immunotherapy

## Abstract

Proteolysis-targeting chimeras (PROTACs) are rapidly emerging as a potential therapeutic strategy for cancer therapy by inducing the degradation of tumor-overexpressing oncogenic proteins. They can specifically catalyze the degradation of target oncogenic proteins by recruiting E3 ligases and utilizing the ubiquitin-proteasome pathway. Since their mode of action is universal, irreversible, recyclable, long-lasting, and applicable to ‘undruggable’ proteins, PROTACs are gradually replacing the role of conventional small molecular inhibitors. Moreover, their application areas are being expanded to cancer immunotherapy as various types of oncogenic proteins that are involved in immunosuppressive tumor microenvironments. However, poor water solubility and low cell permeability considerably restrict the pharmacokinetic (PK) property, which necessitates the use of appropriate delivery systems for cancer immunotherapy. In this review, the general characteristics, developmental status, and PK of PROTACs are first briefly covered. Next, recent studies on the application of various types of passive or active targeting delivery systems for PROTACs are introduced, and their effects on the PK and tumor-targeting ability of PROTACs are described. Finally, recent drug delivery systems of PROTACs for cancer immunotherapy are summarized. The adoption of an adequate delivery system for PROTAC is expected to accelerate the clinical translation of PROTACs, as well as improve its efficacy for cancer therapy.

## 1. Introduction

Oncogenic proteins refer to proteins synthesized by the activation and translation of dysregulated oncogenes, increasing the risk of tumorigenesis [1]. Cancer cells overexpress a variety of oncogenic proteins which are deeply involved in their uncontrolled proliferation and migration; therefore, the suppression of oncogenic protein activity is essential for cancer treatment [2]. Several techniques have been sought to silence oncogenic proteins, including small molecular protein inhibitors, antibodies, small-interfering ribonucleic acids (siRNAs), and clustered regularly interspaced short palindromic repeats (CRISPRs) [3,4,5,6]. They have shown effective protein silencing properties via specifically occupying the active pockets of oncogenic proteins and inactivating them, or knocking down/out oncogenes for protein translation. However, the protein blockade approach using small inhibitors or antibodies exerts some drawbacks, such as demand on strong and specific binding affinity, short and temporal treatment duration, and limited application range [7]. Gene-based therapeutics also bear a couple of constraints in their production and processing cost, transfection efficiency, off-target safety, and vulnerability to enzymatic activity [8,9]. The off-target toxicity of gene-based therapeutics is especially the most critical and chronic issue, as they leave irreversible and permanent damage to normal genes.

In the last two decades, proteolysis-targeting chimeras (PROTACs) were newly introduced as promising strategies for target protein silencing. They are small molecule-based therapeutics with heterobifunctional molecular structures, concurrently binding to target proteins and E3 ligases [10,11]. PROTACs catalyze the degradation of target proteins by employing the E3 ligase and the ubiquitin-proteasome system (UPS), thereby suppressing the activity of proteins in more fundamental aspects [12,13]. Their mode of action (MOA) is completely distinguished from other methods, which brings out unique advantages to themselves. They directly target and degrade oncogenic proteins while negligibly impacting genes, which is preferable to gene-based therapeutics [14]. Compared to inhibitors and antibodies, the therapeutic effect of PROTACs is more persistent as they eliminate oncogenic proteins catalytically and irreversibly [15]. Moreover, their protein degradation is less affected by binding specificity or affinity, making them universally applicable to ‘undruggable’ oncogenic proteins whose tertiary structures are flat and smooth, thereby hard for inhibitors to maintain a constant binding [16,17].

Encouraged by their versatility, PROTACs have been recently investigated for cancer immunotherapy. Cancers are often highly resistant to immune surveillance due to the immunosuppressive tumor microenvironments (TMEs) represented by the overexpression of immune checkpoints, excessive secretion of immunomodulatory cytokines, and prevailing infiltration of immunosuppressive cells, such as tumor-associated macrophages with M2 phenotypes, regulatory T cells (Tregs), and myeloid-derived suppressor cells (MDSCs) [18,19,20]. TMEs restrict the invasion of effective immune cells and hinder their antitumor immunity, whereas oncogenic proteins are directly or indirectly related to establishing those circumstances [21]. Thus, the PROTAC-mediated degradation of oncogenic proteins facilitates the antitumor immune activation through the straightforward disruption of the immune checkpoint system or elimination of immunomodulatory signaling in tumor tissues [22]. In addition, the depletion of certain proteins by PROTAC treatment can induce cancer cell apoptosis that discharges damage-associated molecular patterns (DAMPs), which is called immunogenic cell death (ICD) [23]. DAMPs can initiate the maturation of dendritic cells and subsequent activation of cytotoxic T cells, elevating antitumor immunity. As oncogenic proteins involved in the immunosuppressive properties of TMEs and their specific mechanisms are being continuously identified, the scope of PROTAC application for anticancer immunotherapy is broadening and anticipated to replace conventional inhibitors with superior therapeutic achievements.

Despite the outstanding features of PROTACs for cancer therapy, their transfer to clinical trials has stagnated due to the inherent shortcomings attributed to structural properties. PROTACs contain at least two protein ligands within their structures and therefore unavoidably have high molecular weights of over 700 Da. They also have many polar chemical bonds that increase topological polar surface area (TPSA). The high molecular weight and large TPSA of PROTAC molecules deteriorate their cell permeability and raise the efflux rate by membrane transporters [24,25]. Furthermore, the poor water solubility of PROTACs causes their instability in biological media. The low serum stability and cell permeability of PROTACs impair their PK and bioavailability, eventually leading to undesirable distribution to normal tissues and therapeutic failure. There is a certain limit to completely overcoming the concerning challenges through the structural modification of PROTACs; therefore, the use of an appropriate delivery system is required.

In this review, recent progress in PROTAC technology is disclosed primarily focusing on its delivery systems and achievements in cancer immunotherapy. After describing the structural characteristics and precise MOAs of PROTACs, some undergoing clinical trials are presented as examples to explain their anticancer effects. Then, their physiological stabilities and PK profiles are outlined to clarify the necessity of adequate delivery systems. In the next chapter, the carriers investigated for PROTAC delivery are classified according to their material types and tumor targeting methods, and their results are summarized. Finally, experimental cases using PROTACs and their delivery systems for cancer immunotherapy are listed in detail to discuss their effect on anticancer immunity.

## 2. PROTAC for Anticancer Therapy

### 2.1. Structural Characteristics and MOA of PROTACs

PROTACs are heterobifunctional small molecules comprised of ligands (warheads) for proteins of interest (POIs), recruiters for E3 ligases, and linkers interconnecting them. Their unique molecular structures enable them to form ternary complexes with POIs and E3 ligases. The basic structure and specific MOAs of PROTACs are illustrated in Figure 1. When PROTACs are administered and internalized to cells, they are simultaneously anchored to both POIs and E3 ligases and bring them adjacent to each other [26]. The proximity of E3 ligases to POIs leads to the covalent attachment of multiple ubiquitins on POIs, called polyubiquitination. The polyubiquitinated POIs are then dissociated from the ternary complexes, recognized by proteasome, and finally degraded into amino acids. Upon completing a cycle of UPS-engaged POI degradation, PROTACs recover their free forms to be recycled. The sequential and iterative mechanisms of PROTACs for POI degradation make their therapeutic effects more efficient and persistent.

A substantial number of PROTACs with different POI warheads and E3 recruiters have been designed and their efficacy for anticancer therapy was evaluated. Referring to the PROTAC-DB, a web-based database provided by T. Hou’s research group, 3270 PROTACs in total have been proposed until 2022 [27]. For PROTAC development, 365 warheads, 82 E3 ligase recruiters, and 1501 linkers with diverse characteristics were also screened. Those components were precisely chosen for PROTAC design according to the purpose and application area, playing a vital role in determining the entire performances of PROTACs including the binding affinity to POIs and E3 ligases, POI degrading efficacy, and PK profile. Notably, the intracellular location of POI should also be taken into account to select the optimal E3 ligase recruiter. It was demonstrated that the alteration of the E3 ligase recruiter depending on the subcellular region where POIs are located can enlarge the degrading efficiency of PROTACs [28]. This is mainly attributed to each type of E3 ligase being differently distributed in subcellular regions.

Among the PROTACs discovered so far, 20 of them are currently undergoing clinical trials for anticancer treatment (Table 1) [26,29,30,31]. They respectively target and degrade 11 different oncogenic proteins by recruiting two other types of E3 ligases, cereblon (CRBN) and von Hippel-Lindau (VHL) E3 ligases. In brief, the treatment of PROTACs strictly suppresses tumor growth and progression via silencing oncogenic proteins that are implicated in intra- or intercellular signaling. For example, bromodomain and extra-terminal families (BETs) are epigenetic proteins redundant in many types of cancer cells and they regulate the transcription of oncogenes such as c-Myc and BCL-2. The degradation of BETs with PROTACs prohibits the oncogene expression related to cell proliferation and survival, driving cancer cell apoptosis [32,33]. In contrast, CRBN and VHL are representative E3 ligases widely adopted for PROTAC activity. They are exceptionally enriched in cancer cells compared to other E3 ligases and easy to recruit as many ligands for them have already been reported [10,25]. Although other E3 ligases, such as a cellular inhibitor of apoptosis protein (cIAP) and mouse double minute 2 homolog (MDM2), were studied as well, the majority of PROTACs are still based on CRBN or VHL recruiters. More detailed explanations about the influence of oncogenic protein degradation in tumors and features of E3 ligase have been covered elsewhere [14,34].

### 2.2. Pharmacokinetic Limitations of PROTACs

PROTACs have far distinct physicochemical properties from typical chemo-drugs mainly due to their molecular structures. Since they are bifunctional molecules composed of ligands for both POIs and E3 ligases, PROTACs inevitably possess high molecular weights exceeding 700 Da and multiple numbers of hydrogen-bond donors/acceptors [7,35]. The structural bulkiness and large TPSA cause poor cell permeability of PROTACs by making them hard to diffuse across the cell membrane, while enhancing their efflux through transporters. In addition, PROTACs are sparingly soluble in water due to their high partition coefficient. The low water solubility of PROTACs impedes their stability in biological circumstances, causing rapid clearance in blood circulation and unexpected biodistribution [36,37]. Insufficient intracellular delivery and bioavailability are typical problems faced by a wide variety of PROTACs. Several endeavors have been conducted to alleviate their inferior properties by chemically optimizing the structure of PROTACs. V. G. Klein et al. suggested that the substitution of amide groups between linkers and warheads for esters would reduce the TPSA of PROTACs and improve their cell permeability by lowering the number of intramolecular hydrogen-bond donors [38]. Similarly, the use of bulky linkers with lipophilic side chains or changing E3 ligase recruiters to more hydrophobic ones was also determined to decrease the molecular polarity and increase cell permeability [39,40]. However, TPSA is proportionally related to water solubility, so reducing the molecular polarity to increase cell permeability can adversely affect the water solubility and PK behavior of PROTACs. The structural optimization of PROTAC molecules is also time-consuming and of low applicability since all PROTAC molecules have different physicochemical parameter values. Furthermore, most of the modification methods are aimed to only modulate their cell permeability rather than their biodistribution.

## 3. Cancer-Specific PROTAC Delivery System

Instead of modifying the chemical structure of PROTAC molecules, the employment of delivery systems is considered more efficient because they can conveniently govern the overall PK behaviors of PROTACs toward showing better therapeutic outcomes. As depicted in Figure 2, various types of delivery systems including organic/inorganic nanoparticles, small-molecular targeting moieties, and antibodies have been explored as PROTAC carriers. They adjust the poor water solubility and cell permeability of PROTACs, regulate their PK profiles, and facilitate their selective localization of PROTACs to tumor tissues via passive and/or active targeting methods. In this chapter, the delivery systems for PROTAC are classified according to their tumor-targeting methods and constituent materials, and their effects on PROTAC administration are described in detail.

### 3.1. Nanoparticle-Based Passive Targeting PROTAC Delivery System

Nanoparticles have long been exploited for the modulation of poor water solubilities and PK of drugs. Nanoparticles possess much larger sizes than drug molecules so the solubility of drugs becomes entirely dependent on that of nanoparticles once drugs are loaded into them [41,42,43]. Generally, up to 5% of drugs are loaded into/onto nanoparticles [44]. Nanoparticles can also provide longer blood circulation for drugs since they have higher stability than small drug molecules in physiological conditions [45,46]. Genexol^®^-PM (paclitaxel formulated with polymeric micelle) and Doxil^®^ (liposomal doxorubicin) are representative examples of the nanoparticle-attributable adjustment of drug physicochemical properties [47,48,49,50]. Several nanoparticle-based delivery systems have been proven to be advantageous for the physicochemical property modulation of drugs, thereby obtaining FDA approval [51]. Moreover, the abnormally porous vascular structures and insufficient lymphatic drainage inside tumor tissues enhance the tumoral accumulation of nanoparticles within certain size ranges, which is commonly known as the enhanced permeability and retention (EPR) or passive targeting effect [52,53,54]. Once localized in tumor tissues, nanoparticles are endocytosed by cancer cells via several uptake pathways, such as micropinocytosis, clathrin-dependent endocytosis, caveolin-dependent endocytosis, and clathrin/caveolin-independent endocytosis, discharging free PROTACs intracellularly [55]. The mechanism of nanoparticle endocytosis is differently decided by the size, shape, stiffness, and surface properties of nanoparticles [56]. For example, ligand-functionalized nanoparticles are mainly taken up by cancer cells via clathrin- or caveolin-dependent pathways [57]. Liposomal nanoparticles can also be fused to cell membranes to deliver PROTACs inside the cell [58]. Inspired by the benefits of nanoparticle-based drug delivery systems, several of them previously designed were taken into application to relieve the inferior physiological behaviors of PROTACs.

Based on their major constituting materials, nanoparticles for drug delivery can be broadly classified into organic and inorganic nanoparticles, and organic nanoparticles are further specified as polymeric and lipid nanoparticles. Polymeric nanoparticles are a typical, but practical, drug delivery system. Their physicochemical and pharmacological properties can be adjusted over a wide range depending on the chemical structures of monomers and/or polymeric chains [59]. Drugs can be conveniently loaded into them via physical or chemical interactions, and the surfaces of polymeric nanoparticles are handily modified to endow additional functionalities [60,61]. Over several decades, a large variety of polymeric nanoparticles with high biocompatibility, physiological stability, and multiple functionalities have been designed and explored for drug delivery [62,63,64]. Among them, the block copolymer of poly(ethylene glycol) (PEG) and poly(*D,L*-lactide-*co*-glycolide) (PLGA) is one of the most representative polymeric drug delivery systems, and has been approved by the FDA [65]. The PEG-PLGA copolymers are biocompatible, biodegradable, and able to self-assemble into micelle structures where hydrophobic drugs can be encapsulated in their cores [66,67,68]. A. Sarawat et al. encapsulated bromodomain 4 (BRD4)-degrading PROTACs into PEG-PLGA nanoparticles for the treatment of pancreatic cancer [69,70]. The ARV-825, a PROTAC molecule with BRD4-binding OTX015 warhead and CRBN E3 ligase recruiter, was stably formulated with PEG-PLGA into nanoparticles, which improved the blood half-life and passive tumor targetability of the PROTAC. The PEG-PLGA nanoparticles gradually released ARV-825 inside the cancer cells, inducing BRD4 degradation and subsequent inhibition of undruggable c-Myc transcription. In addition to the PEG-PLGA nanoparticles, several surface-modified polymeric nanoparticles have also been examined for PROTAC delivery, which will be covered in Section 3.2.

Liposomes and lipid nanoparticles are mainly composed of amphiphilic lipids which self-assemble into nanostructures in aqueous media. They have some attractive advantages over polymeric or inorganic nanoparticle-based drug carriers in terms of higher biocompatibility and feasibility of mass production [71]. In addition, lipid nanoparticles can capacitate various types of therapeutic agents in large amounts and their loading method is relatively simple. Several factors determine the sizes and morphologies of lipid nanoparticles, such as the chemical structure and composition of lipids, and their formulation methods [72]. Since the morphological characteristics of lipid nanoparticles can affect their drug loading/releasing profiles, physiological stability, and PK, those factors are precisely controlled according to the properties of payloads [73,74,75].

The studies on lipid-based PROTAC delivery were mainly carried out by K. Patel’s research group. In their first report about PROTAC delivery, Kolliphor ELP^®^, an industrial pharmaceutical surfactant, was newly adopted for the lipid-based nanoformulation of BRD4-degrading ARV-825 and its application to vemurafenib (BRAF inhibitor)-resistant melanoma [36]. The PROTAC nanoformulation was measured to be ~45 nm in its diameter, effectively mitigating the poor water solubility of ARV-825. The nanoformulation showed higher cellular uptake and cytotoxicity against melanoma cells than free PROTAC solutions. Similarly, the PEGylated lipid nanoparticles composed of three different lipid molecules (PEG 2.000-conjugated 1,2-distearoyl-*sn*-glycero-3-phosphoethanolamine; PEG_2,000_-DSPE, PRECIROL^®^ ATO 5, CAPTEX^®^ 300 EP/NF) and two different surfactants (poloxamer 407 and Tween 80) were also evaluated for ARV-825 delivery [76]. The PEGylated lipid nanoparticles were determined to have ~56 nm of hydrodynamic volume and 99.6% of PROTAC loading efficiency. The encapsulation of ARV-825 into PEGylated lipid nanoparticles provided stable and long-lasting dispersity in aqueous media and favorable hemocompatibility. In their most recent study, ARV-825 and nintedanib co-encapsulating liposomes (ARNIPL) were suggested for the combination therapy of vemurafenib-resistant melanoma [77]. PEGylated liposomes with sizes of ~111 nm were formed through the hydration of three lipid mixtures (PEG_2,000_-DSPE, cholesterol, and 1,2-Dioleoyl-sn-glycero-3 phosphocholine; DOPC), and both ARV-825 and nintedanib were loaded in the liposomal bilayer due to their hydrophobicity. The ARNIPL efficiently modulated the poor solubility of both therapeutic agents and secured their stability for a month. After the treatment of ARNIPL to 3D cancer spheroids, ARV-825 induced the BRD4 depletion-associated down-regulation of c-Myc, and nintedanib inhibited the transforming growth factor beta 1 (TGF-β1), resulting in synergistic antitumor efficacy. Throughout the series of investigation by Patel’s group, lipid nanoparticles have demonstrated their remarkable performance for PROTAC delivery.

Apart from this, J. Chen et al. utilized lipid nanoparticles for the intracellular delivery of BRD4-degrading PROTACs pre-fused with E3 ligases [78]. The ARV-771s, which consist of BRD4-binding warheads and VHL E3 ligase recruiters, were preferentially fused with VHL proteins in vitro to prepare pre-fused PROTACs, and they were subsequently encapsulated in 80-O14B lipid nanoparticles (Figure 3A). The pre-fused PROTACs were supposed to increase the BRD4 degrading efficiency by converting the ternary complex-requiring UPS system to a simple binary complex system. Lipid nanoparticles helped pre-fused PROTACs to be internalized to cancer cells, escape from lysosomes, and be smoothly operated inside cytosol.

Inorganic nanoparticles, such as silica, gold, iron oxide, and quantum dot, are other candidates for PROTAC delivery that can exhibit distinguished properties from organic nanoparticles. The morphology and size distribution of inorganic nanoparticles can be controlled in more detail compared to organic nanoparticles, and their rigid structure reduces the risk of unfavorable drug leakage at off-target sites [79]. In particular, gold nanoparticle (GNP) has been widely studied as a drug carrier due to its bioinertness, well-established surface modification method, and versatility to endow additional functionalities. In the study by Y. Wang et al., a GNP-based multi-headed PROTAC platform (Cer/Pom-PEG@GNP) was newly designed for the treatment of non-small-cell lung cancer (NSCLC) (Figure 3B) [80]. Both anaplastic lymphoma kinase (ALK)-binding warheads (Cer-PEG-SH) and CRBN E3 ligase recruiters (Pom-PEG-SH) were conjugated onto GNPs sized to ~32 nm in diameter via the thiol-gold interaction, resulting in ALK-degrading PROTACs. The molar feed ratio of GNP, Cer-PEG-SH, and Pom-PEG-SH for the Cer/Pom-PEG@GNPs synthesis was controlled to 1:50:50. When treated to ALK-positive NSCLC cells, the Cer/Pom-PEG@GNPs effectively degraded ALKs and induced ALK deficiency-related cancer cell death. The multi-headed characteristics of Cer/Pom-PEG@GNPs were to promote the encounter of ALKs and E3 ligases, showing better therapeutic efficacy than small-molecular bifunctional PROTACs. Moreover, the application of GNPs was expected to modulate not only the poor PK of PROTAC, but also its tumor-specific accumulation via the EPR effect. In another study, peptidic PROTACs composed of ligands for MDMX protein and VHL were conjugated to gold(I) via thiol-gold interaction and self-assembled to nanoparticles (Nano-MP@PSIs) for tumor-specific MDMX-degrading PROTAC delivery [81]. Synthesized Nano-MP@PSIs were supposed to release free PROTACs in response to the glutathione-overexpressing tumor microenvironment. After being administered intravenously to pancreatic cancer-xenografted mice, Nano-MP@PSIs exhibited prolonged blood circulation and tumor-selective PROTAC release, and the released free PROTACs subsequently induced MDMX degradation. The degradation of MDMX in cancer cells re-activated the p53 and p73, which reduced anticancer resistance and eventually provoked apoptosis of cancer cells.

Nevertheless, other inorganic nanoparticles besides gold have not yet been explored for PROTAC delivery; thus, further studies need to be arranged.

**Figure 3 pharmaceutics-15-00411-f003:**
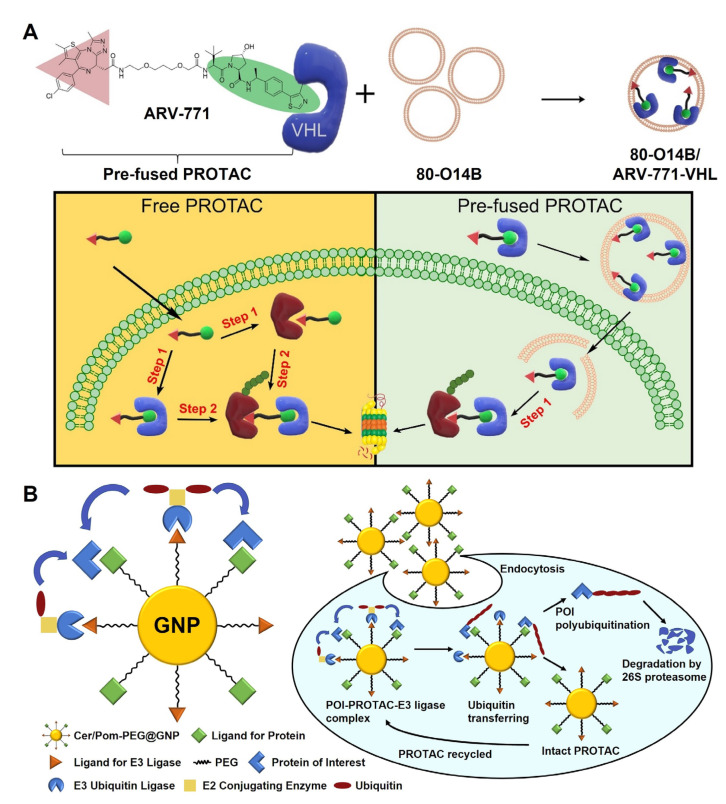
Nanoparticle-based PROTAC delivery. (**A**) Lipid nanoparticle-based PROTAC delivery. BRD4-degrading PROTACs pre-fused with VHL E3 ligases were encapsulated in 80-O14B lipid nanoparticles. The lipid nanoparticle assisted the transfer of pre-fused PROTACs to cytosols, rapidly eliminating intracellular BRD4. Reproduced with permission from [78]. Copyright 2020, Elsevier. (**B**) GNP-based multi-headed PROTAC was prepared by modifying the surfaces of GNPs with both ALK-binding warheads and CRBN recruiters, which increased the chance of encounter between POIs and E3 ligases. Reproduced with permission from [80]. Copyright 2020, Elsevier.

### 3.2. Active Targeting PROTAC Delivery System

#### 3.2.1. Targeting Moiety-PROTAC Conjugates

Active targeting approaches utilize the specific binding affinities of targeting ligands to cancer-overexpressed membrane receptors. Through the utilization of active targeting moieties for PROTAC delivery, the poor cell permeability of PROTAC can be modulated as they promote receptor-mediated endocytosis rather than being internalized through the simple transmembrane diffusion pathway [24,82,83]. The endocytosed PROTACs by active targeting moieties are supposed to be localized in the cytosol, catalyzing the degradation of POIs [84]. Furthermore, the unexpected distribution of PROTAC delivery systems to normal tissues may cause undesirable protein degradation inside normal cells or carrier-associated inflammation [85]. Active targeting moieties promote the cancer-selective accumulation of conjugated PROTACs, reducing their systemic toxicity due to the off-target delivery [86,87]. In general, PROTAC molecules used to be chemically attached to targeting moieties via stimuli-cleavable bonds, so that tumor-internalized targeting moiety-PROTAC conjugates are subsequently cleaved to release free PROTAC molecules. They also elicit prodrug-like characteristics, as PROTACs maintain an inactive state when attached to ligands and restore their activity only after cleavage.

Several active targeting moieties have been examined for PROTAC delivery, which can be classified into small molecules, antibodies, and aptamers depending on their molecular structures. J. Liu et al. adopted small molecular folate groups as targeting ligands for cancer-selective PROTAC delivery (Figure 4A) [88]. Folate groups were conjugated to PROTAC molecules via forming ester groups which were designed to be cleaved by intracellular hydrolases. Since folate receptors are highly expressed in various cancers, such as ovarian, breast, kidney, and colorectal cancers, compared to other normal tissues, the ‘folate-caged’ PROTACs were specifically taken up by cancer cells in a receptor-dependent manner and successfully mediated POI degradation [89,90]. Moreover, ‘folate-caged’ PROTAC was demonstrated with its universality as it was confirmed to be applicable to three different types of PROTACs.

Although small molecular ligands can endow a cancer-specific binding ability to PROTAC molecules, they are not effective enough to modulate the poor water solubility of PROTAC molecules since the molecular sizes of those ligands are too small. In contrast, antibodies, which can specifically target antigens on cancer cell membranes, are proteins with average molecular weights of ~150 kDa, approximately 100 times heavier than usual PROTAC molecules [91]. Due to their relatively bulky structure, antibodies not only compensate for the poor solubility of conjugated PROTACs, but also determine the PK fate [92,93]. In addition to their active tumor targetability, antibodies usually have a longer blood half-life than small molecular ligands, which are more suitable for drug delivery. Taking advantage of antibodies as drug carriers, antibody-drug conjugates (ADCs) have been widely evaluated for anticancer therapy, and as a result, several types of them have been clinically available under FDA approval [94,95,96,97]. Motivated by ADCs, estrogen receptor (ER)-degrading PROTACs were conjugated to human epidermal growth factor receptor-2 (HER2)- or CD22-targeting monoclonal antibodies (mAbs) for the treatment of ER-positive breast cancers [98]. In addition, three different linkages, valine-citrulline, disulfide, and diphosphate, were independently introduced between PROTACs and mAbs to discover the optimal chemical structure of mAb-PROTAC conjugates. All three linkages could be successfully cleaved to recover the free form of ER-degrading PROTACs in response to TMEs, which successively led to ER down-regulation inside cancer cells. Importantly, the preliminary in vivo test demonstrated that the mAb conjugation can considerably improve the stability and PK of PROTACs during their circulation. The same research group also carried out a series of experiments for the design of the most effective mAb-PROTAC conjugates using BRD4-degrading PROTACs [99,100]. They have tried to optimize all the structural components of mAb-PROTACs including the type of stimuli-cleavable linkages between mAb and PROTAC, the structure and length of the PROTAC linker, and the intramolecular location of PROTAC chemically bound to mAbs. All designed mAb-PROTACs were assessed for their PROTAC/mAb ratio, binding affinity to BRD4 and E3 ligase, and IC_50_ values on BRD4 and downstream c-Myc. Their in vitro antiproliferation effect on cancer cells and in vivo antitumor efficacy were evaluated as well, identifying some favorable candidates among them. In another study by M. Maneiro et al., the BRD4-degrading PROTACs were conjugated to anti-HER2 mAbs (trastuzumab) to target HER2-positive breast cancers (Figure 4B) [101]. The mAb-PROTAC conjugates exhibited energetic tumoral uptake due to the specific targetability of trastuzumab, while being rarely internalized to HER2-negative normal cells. After their endocytosis to cancer cells, the bonds between PROTACs and trastuzumabs were hydrolyzed to release free PROTACs, inducing irreversible BRD4 degradation.
Figure 4Active targeting moieties for PROTAC delivery. (**A**) The conjugation of folate molecules to PROTACs enhanced the cellular internalization of PROTACs via folate receptor-related pathways. The folate groups were supposed to be hydrolyzed to release free PROTACs in intracellular conditions. Reproduced with permission from [88]. Copyright 2021, American Chemical Society. (**B**) Trastuzumab was exploited for PROTAC delivery, promoting the selective accumulation of PROTACs in HER2-positive cancer cells. Reproduced with permission from [101]. Copyright 2021, American Chemical Society.
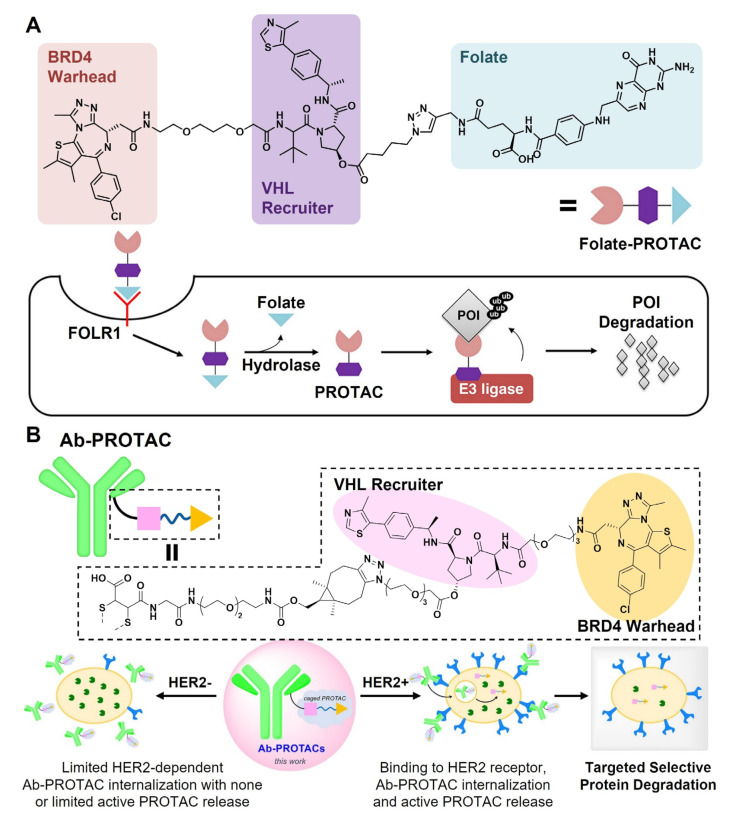


Aptamers are ribonucleic acid (RNA)- or deoxyribonucleic acid (DNA)-based targeting moieties with short and single-stranded primary structures [102,103,104]. By rearranging their nucleic sequences, their 3D-folded structures can be precisely engineered to selectively bind with wide ranges of target membrane proteins. Due to their high binding affinity, stable reproducibility, and low immunogenicity, aptamers have been extensively investigated for molecular imaging and drug delivery [105,106,107,108]. Recently, aptamer-PROTAC conjugates composed of BET-degrading PROTACs and nucleolin-targeting aptamers were newly proposed for the treatment of breast cancers [109]. PROTACs were chemically attached to aptamers via disulfide group-containing short chains which can be cleaved under the presence of glutathione (GSH). The aptamer-PROTAC conjugates showed remarkable BET degradation activity exclusively to nucleolin-overexpressing cancer cells, favorably adjusting the poor solubility and PK profile of native PROTACs as well. Aptamer-PROTAC conjugates were especially expected to be more advantageous over antibody-PROTAC conjugates since they have higher physiological stability and can circumvent the antibody-attributed immunogenicity [110,111]. L. Zhang et al. developed an aptamer-based PROTAC using nucleolin-binding aptamer and VHL E3 ligase recruiter, whose concept was slightly different from that of aptamer-PROTAC conjugates [112]. Aptamers were directly connected to E3 ligase recruiters without any additional POI warheads or stimuli-sensitive bonds, so the aptamer performed as both active targeting moieties and POI ligands. The aptamer-based PROTACs presented much higher solubility and prolonged circulation time compared to conventional small molecule-based PROTACs, and they specifically targeted tumor-overexpressed nucleolin, also catalyzing its degradation. The down-regulation of plasma membrane nucleolin by aptamer-based PROTACs consequently caused the suppression of tumor proliferation and migration [113,114].

#### 3.2.2. Targeting Moiety-Functionalized Nanoparticles

The direct conjugation of PROTAC molecules to antibodies or aptamers has proven to increase the tumor-selective accumulation of PROTACs, as well as successfully modulating their undesirable PK behavior. However, mAb-PROTAC has a limitation in its loading capacity. The drug/antibody ratio of conventional ADCs is controlled not to exceed four, since the overloading of drug molecules onto antibodies may cause their aggregation or rapid clearance in physiological conditions, and finally diminish their therapeutic efficacy [53,115,116]. Taking into account that the average molecular weight and hydrophobicity of PROTACs are higher than those of chemo-drugs, the PROTAC/antibody ratio should be lower than four. As an alternative, active targeting moiety-functionalized nanoparticles can be used as PROTAC carriers. Nanoparticles are generally less affected in their PK by the loading amount than antibodies, thereby being able to capacitate a relatively large amount of PROTACs. Moreover, the surface modification of nanoparticles with active targeting moieties can facilitate the enhanced tumor accumulation of PROTACs by exploiting both passive and active targeting methods.

The first case of exploiting active targeting moiety-functionalized polymeric nanoparticles for PROTAC delivery was reported in 2020, which was composed of poly(*D,L*-lactide) (PLA) nanoparticles coated with polyethyleneimine (PEI) [117]. The BRD4-degrading PROTAC (MZ1), consisting of BRD4-binding ligand and VHL E3 ligase recruiter, was loaded in PLA nanoparticles via the nanoprecipitation method, and trastuzumab was conjugated onto the PEI layer, being formulated with a size of ~114 nm. The loading amount of MZ1 in the nanoparticle was measured as 0.5%, and the loaded MZ1 was steadily released in aqueous media. The trastuzumab on the particle surface notably enhanced the internalization of the nanoparticles to HER2-expressing cancer cells, and the endocytosed MZ1 brought into their BRD4 deficiency-associated apoptosis. Afterward, a study about the combination therapy with PROTAC and chemo-drug was addressed using an active targetable polymeric nanoparticle carrier [118]. The cyclo(Arg-Gly-Asp-d-Phy-Lys)s (cRGDfk) were conjugated to PEG-PLA block copolymer to endow specific binding to α_v_β_3_ integrin, and the cRGD-PEG-PLAs spontaneously assembled with PEG-PLAs into nanoparticles of ~40 nm. BRD4-degrading ARV-825 and doxorubicin were encapsulated together in the nanoparticles and the resulting formulation was applied to treat glioma. The cRGD sequences on the nanoparticle surfaces facilitated the co-delivery of ARV-825 and doxorubicin into glioma cells. Moreover, doxorubicin and ARV-825 elicited synergistic anticancer effects since the ARV-825 significantly relieved the resistance of tumors against doxorubicin by depleting the overexpressed BRD4. Further, J. Gao et al. suggested a multifunctional polymeric delivery system for PROTAC-based combination cancer therapy (Figure 5) [119]. A self-assembled polymeric nanoparticle was designed to be directed to tumor tissues via bioorthogonal chemistry, and PEGs on their surfaces were to be cleaved in response to matrix metalloproteinase 2 (MMP2) in TMEs. The BRD4-degrading ARV-771 or MZ1 and photosensitizer were chemically conjugated to the core block of polymer nanoparticles with glutathione (GSH)-sensitive bonds. The proposed polymeric nanoparticles enabled a highly selective delivery of PROTAC and photosensitizer to tumor tissues by adapting both bioorthogonal chemistry and multiple stimuli-sensitive strategies. When they were administered to breast cancer-bearing models, the BRD4 degradation and photodynamic therapy (PDT) cooperatively boosted the apoptosis of cancer cells, thereby achieving efficient tumor regression.

Lipid nanoparticles are another PROTAC carrier where additional moieties can be introduced for active cancer targeting. A. Saraswat et al. explored galactose-decorated liposomes for PROTAC delivery to treat hepatocellular carcinoma [120]. The galactose on liposome surfaces was intended to specifically bind asialoglycoprotein receptors (ASGPRs) on cancer cells, and BRD4-degrading ARV-825 was loaded in the bilayer of liposomes to produce GALARV. GALARV showed an improved delivery efficiency compared to liposomes without galactose or free ARV-825, and it significantly reduced the intracellular level of BRD4 and downstream c-Myc and BCL-2. In a 3D tumor spheroid model, GALARV was confirmed to effectively inhibit its proliferation by inducing apoptosis deep inside the spheroid. The cell-derived biomimetic lipid nanoparticles were also examined for targeted PROTAC delivery against pancreatic cancers [121]. The biomimetic nanoparticles were extracted from homologous cancer cells, thereby readily targeting tumor tissues without further modification. Due to the cell membrane-attributed camouflage effect, they also displayed enhanced stability and compatibility in physiological conditions. Phosphodiesterase δ (PDEδ)-degrading PROTACs (PIPDs) which recruit CRBN E3 ligases were encapsulated to the biomimetic nanoparticles, formulated as ~125 nm in diameter. PIPDs released from nanoparticles remarkably degraded PDEδs inside pancreatic cancer cells, and the following RAS signaling inhibition induced their apoptosis at last.

## 4. PROTACs and Their Delivery System for Cancer Immunotherapy

### 4.1. PROTACs for Cancer Immunotherapy

Oncogenic proteins not only act as essential factors for the growth, proliferation, and metastasis of cancer cells, but are deeply engaged in constructing immunosuppressive TMEs as well. They assist cancer cells to avoid being recognized by cytotoxic T cells, and express signals to gather regulatory immune cells to tumor tissues [122,123,124]. In this regard, some PROTACs have recently been investigated for cancer immunotherapy, as they can activate anticancer immunity by degrading oncogenic proteins. The treatment of PROTACs eliminates the immunosuppressive proteins or induces protein deficiency-related cancer ICD, converting ‘cold tumors’ into ‘hot tumors’ (Figure 6). The PROTACs examined for cancer immunotherapy are listed in Table 2 according to their target proteins, and their therapeutic outcomes are summarized in brief.

Programmed death ligand 1 (PD-L1) is a type of membrane protein called immune checkpoint which can interact with receptor programmed death 1 (PD-1) on cytotoxic T cells and inactivate them [125]. PD-L1 was originally to prevent the autoimmune attack of T cells on normal tissues, but considerable types of cancer cells have also been known to overexpress PD-L1 to evade anticancer immunity [126,127]. Therefore, the inhibition or degradation of PD-L1s on cancer cells is essential to regulate the immunosuppressive circumstances of tumor tissues. B. Cheng et al. designed the first PD-L1-degrading PROTACs using resorcinol diphenyl ether-based PD-L1 ligands and CRBN E3 ligase recruiters [128]. Several candidates with different structures of PD-L1 ligands and linkers are examined against PD-L1-overexpressing human breast cancer cells, successfully discovering the optimal PD-L1-degrading PROTAC molecule (P22). The P22 effectively induced the PD-L1 degradation with a low IC_50_ value and its MOA was found to depend on the lysosome-associated pathway. Moreover, P22 treatment stimulated cancer cells to release interferon-γ (IFN-γ) which led to the activation of co-cultured CD3 T cells. Another PD-L1-degrading PROTAC was suggested by Y. Wang et al., which was composed of a small molecular PD-L1 inhibitor (BMS-37) and CRBN E3 ligase recruiter [129]. After identifying the favorable linker structure, the PD-L1-degrading PROTAC was confirmed with its efficacy in several types of human hematological malignant cells. When it was administered to colon cancer-bearing mice, the tumor growth was notably suppressed as it eradicated intratumoral PD-L1s and promoted the tumor invasion of cytotoxic T cells. In a study by Y. Liu et al., it was further attempted to find the most suitable E3 ligase recruiter and PD-L1 ligand for PD-L1-degrading PROTAC [130]. Four different recruiters for VHL, CRBN, MDM2, and cIAP E3 ligase and four PD-L1 ligands including BMS-37 were combinatorically selected and conjugated to each other, and the resulting PROTACs were examined for their effect against melanoma cells in vitro. Among them, the PROTAC molecule composed of CRBN E3 ligase recruiter, BMS-37, and C3 linker (BMS-37-C3) was determined to have the highest efficiency. BMS-37-C3 showed strong PD-L1 binding affinity, active PD-L1 degrading ability, and following remarkable T cell activation. In addition to the small molecule-based PROTACs, a novel antibody-based PROTAC (AbTAC) has been reported as well for PD-L1 degradation [131]. On the basis of bispecific recombinant immunoglobulin G, AbTAC was precisely tailored to bind to both PD-L1 and RNF43 transmembrane E3 ligase simultaneously. The AbTAC induced the internalization of PD-L1 and RNF43 at once, causing the successful lysosomal degradation of PD-L1. The AbTAC could not catalyze the PD-L1 degradation in RNF43 knockdown or lysosome-inhibited cells, confirming that its MOA is closely associated with both RNF43 and lysosomal pathways.

Indoleamine 2,3-dioxygenase (IDO) plays an important role in cancer-related immunomodulation [132]. It is overexpressed in diverse types of cancers such as bladder, lung, kidney, and breast cancer, and it catalyzes the metabolism of tryptophan to kynurenine [133,134,135,136]. The tryptophan insufficiency in the tumor microenvironment decreases the activity of cytotoxic T cells, and the abundance of kynurenine promotes the proliferation of Tregs and the conversion of macrophages into immunosuppressive phenotypes [137,138]. M. Hu et al. developed the first IDO-targeting PROTAC molecule and examined its immune-activating property [139]. The PROTAC was designed to contain an Epacadostat-derived IDO warhead and a CRBN E3 ligase recruiter, which specifically degraded IDOs in HeLa cells. Although it did not elicit much cytotoxicity in cancer cells by itself, the IDO-targeting PROTAC effectively induced kynurenine depletion and the following killing activity of chimeric antigen receptor-T (CAR-T) cells.

Epidermal growth factor receptor (EGFR) is well-known as a key oncogenic protein in non-small cell lung cancer (NSCLC) [140]. It is aggressively expressed and mutated in cancer cells, boosting their proliferation and drug resistance [141,142]. Moreover, it promotes the overexpression of both IDO and PD-L1 which provide strong immunosuppressive properties to cancer cells [143,144]. Especially, EGFR-targeting anticancer therapies using conventional inhibitors or antibodies failed often due to the side effect of therapeutic agents and the mutation of EGFR. K. Wang et al. proposed an EGFR-targeting PROTAC composed of gefitinib (warhead for EGFR with L858R mutation) and VHL E3 ligase recruiter [145]. The EGFR-targeting PROTAC not only permanently degraded EGFR, but also successfully inhibited the downstream expression of PD-L1 and IDO. After being treated to NSCLC tumor-bearing in vivo models, the EGFR-targeting PROTAC exhibited a better tumor growth suppression effect than gefitinib, promoting the migration of CD3-positive T cells to tumor tissues.

The BET is one type of oncogenic protein that has been extensively studied as a target of PROTAC-mediated degradation for cancer therapy; therefore, some BET-degrading PROTACs are undergoing clinical trials [146]. In addition to their tumor proliferation inhibitory and cancer cell apoptotic effects, BET-degrading PROTACs were determined to induce ICD and activate immune reactions against tumor tissues [147]. When BETd260, a PROTAC molecule consisting of HJB97 (warhead for BET) and thalidomide, was used to treat colorectal cancers, it stimulated the expression of death receptor 5 (DR5) and BET deficiency-mediated cell apoptosis. The increased release of calreticulin (CRT), which is a type of DAMPs, was observed during the cancer cell apoptosis in a DR5-dependent manner, demonstrating its ICD effect. Furthermore, the combination therapy of BETd260 and anti-PD-L1 exhibited notable antitumor efficacy with high infiltration of cytotoxic T lymphocytes into tumor tissues. In another study by S. M. Jensen et al., BET-targeting PROTAC was reported to further lead to present specific class-I major histocompatibility complex (MHC-I) peptides [148]. Against chronic myeloid leukemia (CML) cells, the BET-targeting PROTACs provoked BET degradation and MHC-I peptides were produced as byproducts. The resulting MHC-I peptides were transported to cell surfaces and recognized by cytotoxic T cells, causing anticancer immune activation [149]. Aside from that, the silencing of BET proteins was also known to decrease PD-L1 expression, which has not been demonstrated with BET-degrading PROTACs yet [150,151].

Nicotinamide phosphoribosyl transferase (NAMPT) is an enzymatic protein that catalyzes the conversion of nicotinamide (NAM) to nicotinamide adenine dinucleotide (NAD^+^) in the energy metabolism pathway [152]. It is often prevalently expressed inside malignant tumors which consistently consume energy sources for their proliferation, and the extracellular NAMPT secreted from malignancies acts like a cytokine that promotes MDSCs to be expanded [153,154]. Y. Wu et al. examined intracellular NAMPT-degrading PROTACs (A7) for the immunotherapy of colon cancers [155]. The administration of A7 effectively degraded the intracellular NAMPT by utilizing VHL E3 ligase, followed by a decrease in its extracellular concentration. In turn, it diminished the tumor infiltration of MDSCs and successfully promoted the immune activation in TMEs.

The B cell lymphoma-extra large (BCL-X_L_) protein is a member of the anti-apoptotic BCL-2 families which is responsible for signals opposing the stress conditions in cancer cells [156,157]. Anti-apoptotic BCL-2 families are most widely recognized for their multidrug resistance-inducing ability; at the same time, they are closely associated with tumor-infiltrating Tregs. It was demonstrated that the treatment of BCL-X_L_-degrading PROTACs (DT2216s) causes the apoptosis of tumor-infiltrating Tregs and encourages the activation of cytotoxic T cells [158]. DT2216s were developed using BCL-X_L_ inhibitors (ABT263) and VHL E3 ligase recruiters, and they could selectively mediate the degradation of BCL-X_L_ in Tregs. They efficiently promoted tumor regression in syngeneic animal tumor models, but did not show meaningful differences in immunodeficient ones, demonstrating the BCL-X_L_ degradation-related anticancer immune activation. Furthermore, due to their selective activity, DT2216 did not exert any serious side toxicity, such as thrombocytopenia which is a major adverse effect of conventional BCL-X_L_ inhibitor.

### 4.2. PROTAC Delivery Systems for Cancer Immunotherapy

The adoption of drug delivery systems for PROTAC-mediated cancer immunotherapy can be a highly advantageous approach since it not only adjusts the poor bioavailability of PROTACs, but enables combinatorial immunotherapy as well. The combination therapy of PROTACs and other treatment methods, such as chemotherapy, photothermal therapy (PTT), and PDT, synergistically stimulate immune responses against cancers, enhancing and prolonging their therapeutic effects. In 2021, K. Pu and his co-workers developed the semiconducting polymer nano-PROTACs (SPN_pro_s) for combinatorial cancer immunotherapy, which was the first report of PROTAC delivery system-based cancer immunotherapy [159]. SPN_pro_s were manufactured by the self-assembly of amphiphilic copolymers composed of hydrophobic semiconducting polymer chains for PDT and hydrophilic PEG brushes. Indoleamine 2,3-dioxygenase (IDO)-degrading PROTACs were chemically conjugated at the end of PEG chains using cathepsin B-responsive peptides (Figure 7). When SPN_pro_s were delivered to tumor tissues, the peptides were specifically cleaved by tumor-overexpressed cathepsin B to release free IDO-degrading PROTACs and the free PROTACs mediated IDO degradation utilizing the VHL E3 ligase-UPS pathway. The irreversible IDO deficiency by SPN_pro_s led to a decreased metabolism of tryptophan to kynurenine and a subsequent reversion of the tumor immunosuppressive environment. Moreover, the semiconducting polymers in SPN_pros_ generated reactive oxygen species (ROS) under the irradiation of near-infrared (NIR) which induced immunogenic cell death (ICD) of cancer cells, resulting in the activation of antitumor immune responses. The combinational therapy using SPN_pro_s successfully inhibited tumor growth and metastasis by boosting antitumor immunity. This PROTAC delivery system has been exploited in another research by K. Pu’s group [160]. The cyclooxygenase 1/2 (COX-1/2)-degrading PROTACs were introduced onto the semiconductor polymer nanoparticles instead of IDO-degrading PROTACs. Similar to the previous study, COX-1/2-degrading PROTACs were specifically released in a cathepsin B-dependent manner, and the released PROTACs provoked COX-1/2 deficiency-associated prostaglandin E_2_ depletion, followed by the alteration of tumor immunosuppressive environments. The immunomodulation effect of COX-1/2-degrading PROTACs was synergistically operated with PDT-mediated ICD, exhibiting efficient anticancer efficacy. In contrast, T. Yang et al. proposed a simple active targeting polymeric nanoparticle for BRD4-degrading PROTAC delivery against glioma [161]. Substance P peptides (SPs), targeting moieties for neurokinin 1 receptors (NK-1Rs), were conjugated to PEG-PLA diblock copolymers and formulated with unmodified PEG-PLA and ARV-825 to prepare polymeric micelles (SPP-ARV-825). SPP-ARV-825s presented a strong BRD4 depletion-related antiproliferation effect, especially on NK-1R-overexpressing glioma cells. Noteworthily, the degradation of BRD4 by ARV-825 not only induced the apoptosis of glioma cells, but also suppressed the tumor-infiltrated M2 macrophage via inhibiting the transcription of downstream IRF4 promoter and the phosphorylation of STAT6, STAT3, and AKT. Therefore, it was confirmed that the BRD4-degrading PROTAC can modulate the M2 macrophage-associated immunosuppressive TMEs.

To date, there are only three reports regarding cancer immunotherapy using the PROTAC delivery system, which have been described above. A majority of research is still focused on the screening of novel POI and/or E3 ligase ligands for PROTACs, and those for the development of the PROTAC delivery system and its application to cancer immunotherapy have only entered the initial stage. Since PROTACs have been proven to have superior effects over conventional inhibitors, they are expected to gradually expand their application range and replace the role of inhibitors in cancer immunotherapy.
pharmaceutics-15-00411-t002_Table 2Table 2Research cases of PROTAC application for cancer immunotherapy.POIE3 LigaseResultsRef.PD-L1CRBNImmune checkpoint degradation,cytotoxic T cell activation[128]CRBN[129]CRBN, MDM2, cIAP, VHL[130]RNF43[131]IDOCRBNTryptophan metabolism inhibition,Treg/M2 macrophage inhibition[139]VHLEGFRVHLDepletion of IDO and PD-L1[145]BETCRBNICD generation[147]CRBN, MDM2, VHLMHC-I peptide expression, cytotoxic T cell activation[148]NAMPTVHLNAM catabolism inhibition, MDSC expansion inhibition[155]BCL-X_L_VHLTreg inhibition, cytotoxic T cell activation[158]COX-1/2VHLPGE2 depletion, Treg/M2 macrophage/MDSC inhibition[160]BRD4CRBNM2 macrophage inhibition[161]
Figure 7PROTAC delivery systems for cancer immunotherapy. The combinational cancer immunotherapy was carried out using semiconducting polymer nano-PROTACs. The semiconducting polymers in nano-PROTACs induced PDT-mediated ICD. IDO-degrading PROTACs were cathepsin B-specifically released from nanoparticles and inhibited the Tryptophan metabolism, reprogramming the immunosuppressive TMEs. Reproduced with permission [159]. Copyright 2021, Nature Publication Group.
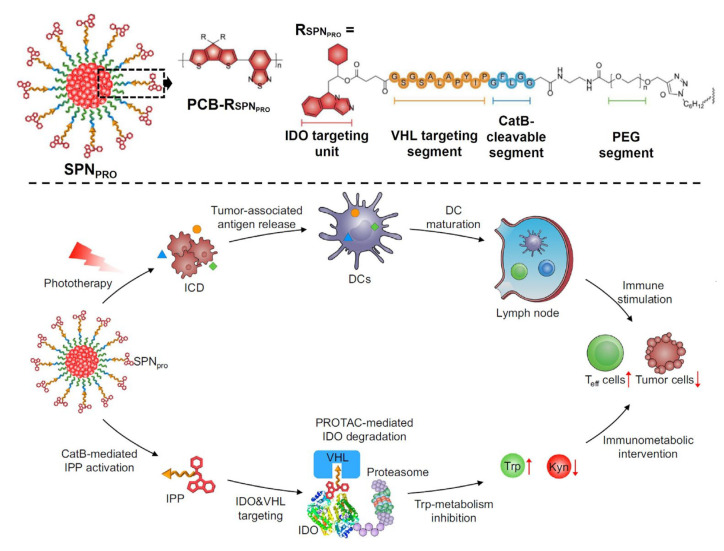


## 5. Conclusions

Herein, the research on the development of PROTAC delivery systems and their application to anticancer therapy and immunotherapy was summarized from the past to the present. The general characteristics and MOAs of PROTACs were priorly noted and their PK limitations were pointed out. Subsequently, various PROTAC delivery systems for cancer therapy were classified depending on their material types, and their effects on modulating the PK of PROTAC were explained. Finally, recent studies exploring PROTACs and their delivery systems for cancer immunotherapy were introduced. Although PROTACs are gradually recognized as potential alternatives for conventional inhibitors, the research outcomes utilizing them are still insufficient. The PROTAC is considered a comparatively novel therapeutic agent whose concept emerged only two decades ago; therefore, its delivery system also has considerable points to be optimized according to the characteristics of PROTAC. Continuous studies on the development of an appropriate delivery system for PROTAC are required to enhance the therapeutic effect of PROTAC-based cancer immunotherapy and further successful clinical translation of PROTAC.

## Figures and Tables

**Figure 1 pharmaceutics-15-00411-f001:**
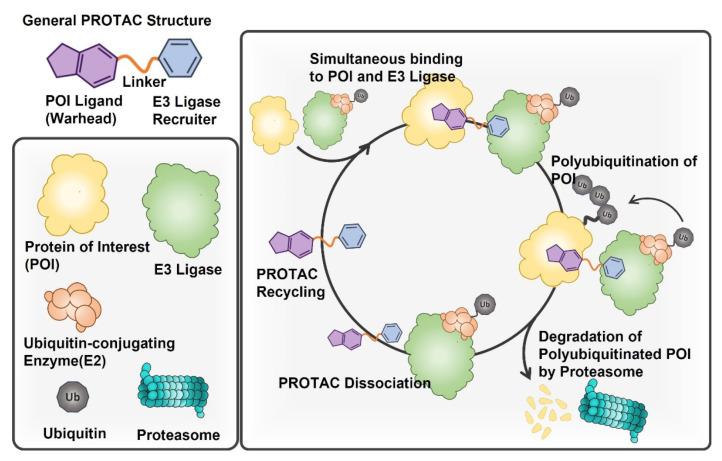
Structure and specific MOA of PROTACs. PROTAC has a structure in which two ligands for E3 ligases and POI are coupled to each other through a linker. It simultaneously anchors both POIs and E3 ligases, letting them approach each other. E3 ligases form a complex with the E2 enzyme and induce the polyubiquitination of adjacent POIs. Proteasome recognizes polyubiquitinated POIs and degrades them into amino acids. The dissociated PROTAC molecule can participate in another degradation cycle of POI.

**Figure 2 pharmaceutics-15-00411-f002:**
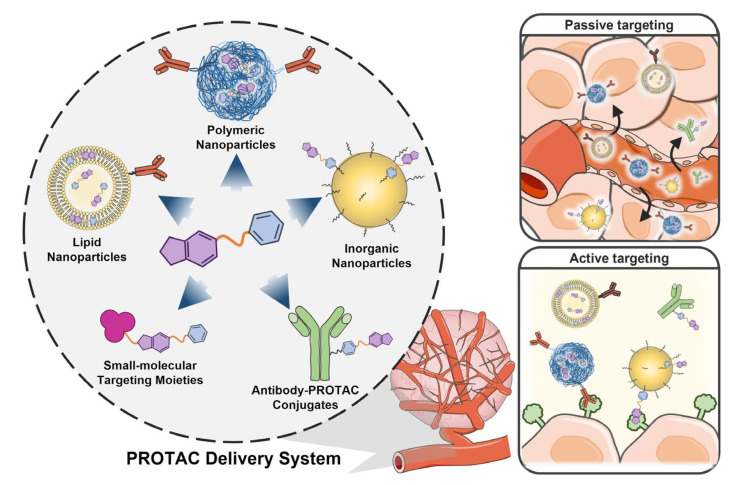
Carriers for PROTAC delivery. Various types of carriers including organic/inorganic nanoparticles, small-molecular targeting moieties, and antibodies/aptamers have been explored for PROTAC delivery. They regulate the pharmacokinetic properties of PROTACs, concurrently increasing their tumor-specific accumulation levels through passive and/or active targeting methods.

**Figure 5 pharmaceutics-15-00411-f005:**
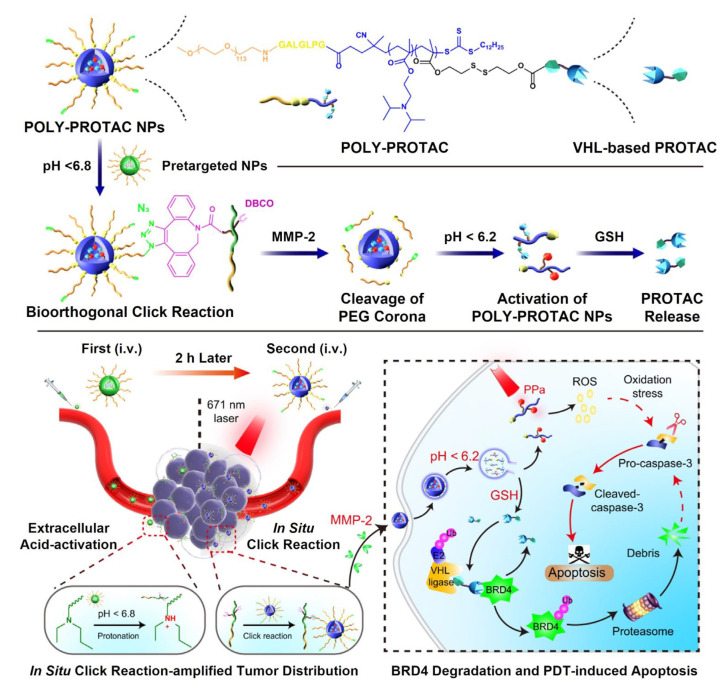
Active targeting nanoparticle-based PROTAC delivery. A multifunctional polymeric nanoparticle was proposed for combination cancer therapy through the co-delivery of PROTAC and photosensitizer. The polymeric nanoparticle was designed to be specifically delivered to tumors via bioorthogonal chemistry and to discharge free PROTACs in response to MMP-2, acidic pH, and GSH. The PROTAC and photosensitizer co-delivered to tumor tissues cooperatively accelerated the cancer apoptosis. Reproduced with permission from [119]. Copyright 2022, Nature Publishing Group.

**Figure 6 pharmaceutics-15-00411-f006:**
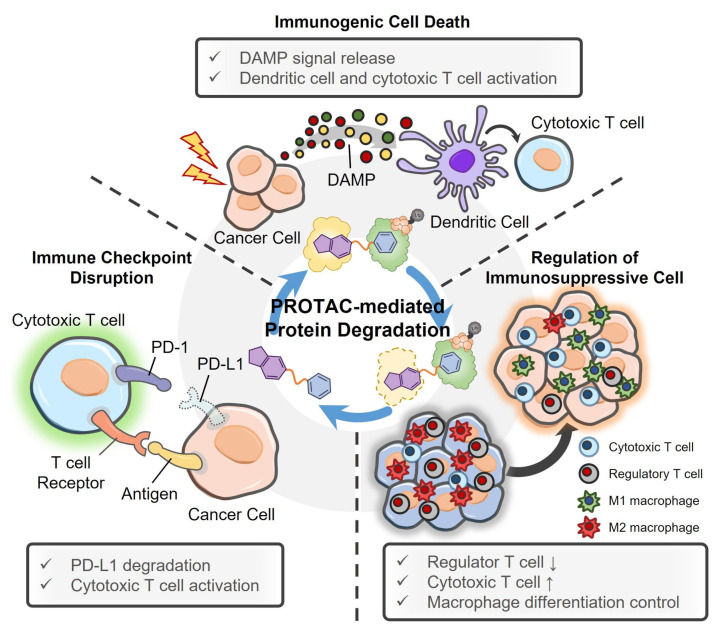
PROTAC-based cancer immunotherapy. The PROTAC treatment can modulate the immunosuppressive tumor microenvironment to be immunoactive via three pathways. First, PROTACs eliminate oncogenic proteins essential for the growth and survival of cancer cells, causing their immunogenic cell death. Second, PROTACs disrupt the immune checkpoint on cancer cells and make them vulnerable to immune attack by cytotoxic T cells. Finally, PROTACs eradicate immunosuppressive signal-associated cytokines, thereby reducing the population of regulatory immune cells in tumor tissues.

**Table 1 pharmaceutics-15-00411-t001:** PROTACs in clinical trials for anticancer therapy.

POI	Name	E3 Ligase	Indication	Clinical Phase	Ref.
Androgen receptor(AR)	AC176	N/A	Prostate cancer	Phase I	NCT05241613
	ARV-110	CRBN		Phase II	NCT03888612
	ARV-766	VHL		Phase II	NCT05067140
	CC-94676	CRBN		Phase I	NCT04428788
	HP518	N/A		Phase I	NCT05252364
B-cell lymphoma-extra large (BCL-X_L_)	DT2216	VHL	Solid tumor, Hematologic malignancy	Phase I	NCT04886622
Bromodomain 4 (BRD4)	RNK05047	N/A	Advanced solid tumor, Diffuse large B cell lymphoma (DLBCL)	Phase I/II	NCT05487170
Bromodomain 9 (BRD9)	CFT8634	CRBN	Synovial sarcoma, Soft tissue sarcoma	Phase I/II	NCT05355753
	FHD-609	CRBN		Phase I	NCT04965753
Bruton’s tyrosine kinase (BTK)	BGB-16673	N/A	B cell malignancies	Phase I	NCT05006716
	HSK29116	CRBN		Phase I	NCT04861779
	NX-2127	CRBN		Phase I	NCT04830137
	NX-5948	CRBN		Phase I	NCT05131022
Epidermal growth factor receptor (EGFR)	CFT8919	CRBN	Non-small cell lung cancer (NSCLC)	IND-e	N/A
Estrogen receptor (ER)	ARV-471	CRBN	Breast cancer	Phase II	NCT04072952
	AC682	CRBN		Phase I	NCT05080842
Interleukin-1 receptor-associated kinase 4 (IRAK4)	KT-413	CRBN	DLBCL	Phase I	NCT05233033
KRAS-G12D	ASP3082	N/A	Solid tumor	Phase I	NCT05382559
Signal transducer and activator of transcription 3 (STAT3)	KT-333	N/A	Liquid and solid tumors	Phase I	NCT05225584
Tyrosine receptor kinase (TRK)	CG001419	CRBN	Cancers and other indications	IND-e	N/A

N/A: Not available.

## Data Availability

Not applicable.

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
