# Peer review of "Cancer-Specific Delivery of Proteolysis-Targeting Chimeras (PROTACs) and Their Application to Cancer Immunotherapy"

_pharmaceutics, 2023, doi:10.3390/pharmaceutics15020411_

Round 1

Reviewer 1 Report

This review article from Prof Kim's group summarizes the PROTAC and its various delivery approaches for anticancer therapy and immunotherapy.

First, this article is well-written and communicates the science behind PROTAC and the challenges in using them as treatment options in a comprehensible way. I enjoyed reading the paper and it has cited all the key important papers in the field.

Second, the presentation of the paper is good. I like the elegance of the illustrations and the clinical trial details that went into the table. I find this information highly relevant and useful for the readers.

Third, although there are multiple review articles about PROTACs, this work emphasizes its significance in immunotherapy with key citations. In that way, this paper brings a lot of value and can be perceived as an interesting paper.

I would request the authors to add some information for the following points.

a)  what is the cellular mechanism associated with the uptake of the nanoparticles based delivery system? Please speculate and cite relevant articles.

b) what is the max-min amount/concentration of PROTAC that can be loaded into different delivery systems? please give a range.

c) what are the off-target effects of this delivery system? where does the nanoparticle localize in general? can we tailor the targeting to a specific location say nucleus or cytoplasm?

d) can we tailor the nanoparticle only for the cancer cells? if so, how?

e) General curiosity question, are there any FDA-approved oncology-based drugs that utilize any of these highlighted nanoparticle-based delivery systems? Please add them to the discussion.

f) how economical or easier it is to achieve uniform stoichiometry of  POI and E3 ligase recruitment in a given GNP-based nanoparticle delivery system?

I enjoyed reading this work and I find it highly relevant and useful to the protein degradation field. I recommend this review for publication with minor changes. Congratulations to the authors.

Author Response

The authors appreciated the reviewer’s valuable comments. After a careful examination of those comments, we have prepared the revised manuscript to reflect them. Our responses to the reviewer’s comments are summarized in the response letter in detail. Please see the attachment.

Reviewer 2 Report

Reviewers’ comments for the Manuscript ID: pharmaceutics-2140653

The manuscript title:Cancer-specific delivery of proteolysis-targeting chimeras 2 (PROTACs) and their application to cancer immunotherapy”.

In the current manuscript, authors comprehensively reviewed different approached to deliver PROTACs specifically to cancer namely passive approach, ex. polymeric or lipophilic nanomaterial, targeting approach which involves small molecular targeting (folate, HER2), antibody-drug conjugates (ADCs), aptamers and other functionalized nano particle.

It is well organized & written manuscript and should be suitable for publication in pharmaceutics in its current forms.

General suggestions

·       Authors ignored citations of some of recently published similar review articles, so they so should be cited before publication.

·       Ex:  doi: 10.3390/molecules27248828,

  • DOI: 10.1186/s12943-021-01434-
  • Abbreviation for MOA should be given
  • In line 212 Enhanced permeability and retention (EPR) should be added.
  • In line 342 folate expression in healthy tissues is “very less or not” should be corrected, because folate receptor is highly over expressed in healthy kidneys.

Author Response

(The authors gave the same response as above.)

Reviewer 3 Report

The review by Y.Moon et al. is a high quality comprehensive study of the state-of-the-art in basic and cancer-related PROTAC technology. The authors have done a great job analyzing a vast corpus of literature including the most recent (2022) publications. The work will be appreciated by the research community due to the detailed analysis as well as to the profound and clear-cut manner of presentation. 

My only recommendation is to avoid a few trivial sentences in Intro about oncoproteins and minimize general statements in regard to cancer biology. This polishing would better focus the text on its specific point. Plus I think mentioning cancer immunotherapy in the title may not be necessary since this aspect is within general scope of the study, that is, PROTAC mediated therapeutic oncoprotein elimination. The authors righly placed it in a separate Section 4, so this problem is duly addressed. Anyway, the decision is at the authors' discretion.       

Author Response

(The authors gave the same response as above.)

Reviewer 4 Report

Kim et al. covered the most recent PROTAC drug delivery systems for cancer immunotherapy in this review article. It is a well-written and well-designed document that will help researchers understand appropriate delivery systems for cancer immunotherapy. I recommend that this review be published in Pharmaceutics in its current form.

Author Response

(The authors gave the same response as above.)
